# Depth-Limited Solving for Imperfect-Information Games

**Noam Brown, Tuomas Sandholm, Brandon Amos**
Computer Science Department
Carnegie Mellon University
noamb@cs.cmu.edu, sandholm@cs.cmu.edu, bamos@cs.cmu.edu

## Abstract

A fundamental challenge in imperfect-information games is that states do not have well-defined values. As a result, depth-limited search algorithms used in single-agent settings and perfect-information games do not apply. This paper introduces a principled way to conduct depth-limited solving in imperfect-information games by allowing the opponent to choose among a number of strategies for the remainder of the game at the depth limit. Each one of these strategies results in a different set of values for leaf nodes. This forces an agent to be robust to the different strategies an opponent may employ. We demonstrate the effectiveness of this approach by building a master-level heads-up no-limit Texas hold'em poker AI that defeats two prior top agents using only a 4-core CPU and 16 GB of memory. Developing such a powerful agent would have previously required a supercomputer.

## 1 Introduction

Imperfect-information games model strategic interactions between agents with hidden information. The primary benchmark for this class of games is poker, specifically heads-up no-limit Texas hold'em (HUNL), in which *Libratus* defeated top humans in 2017 [6]. The key breakthrough that led to superhuman performance was nested solving, in which the agent repeatedly calculates a finer-grained strategy in real time (for just a portion of the full game) as play proceeds down the game tree [5, 27, 6].

However, real-time subgame solving was too expensive for *Libratus* in the first half of the game because the portion of the game tree *Libratus* solved in real time, known as the *subgame*, always extended to the end of the game. Instead, for the first half of the game *Libratus* pre-computed a fine-grained strategy that was used as a lookup table. While this pre-computed strategy was successful, it required millions of core hours and terabytes of memory to calculate. Moreover, in deeper sequential games the computational cost of this approach would be even more expensive because either longer subgames or a larger pre-computed strategy would need to be solved. A more general approach would be to solve *depth-limited subgames*, which may not extend to the end of the game. These could be solved even in the early portions of a game.

The poker AI *DeepStack* does this using a technique similar to nested solving that was developed independently [27]. However, while *DeepStack* defeated a set of non-elite human professionals in HUNL, it never defeated prior top AIs despite using over one million core hours to train the agent, suggesting its approach may not be sufficiently efficient in domains like poker. We discuss this in more detail in Section 7. This paper introduces a different approach to depth-limited solving that defeats prior top AIs and is computationally orders of magnitude less expensive.

When conducting depth-limited solving, a primary challenge is determining what values to substitute at the leaf nodes of the depth-limited subgame. In perfect-information depth-limited subgames, the value substituted at leaf nodes is simply an estimate of the state's value when all players play an

equilibrium [35, 33]. For example, this approach was used to achieve superhuman performance in backgammon [39], chess [9], and Go [36, 37]. The same approach is also widely used in single-agent settings such as heuristic search [30, 24, 31, 15]. Indeed, in single-agent and perfect-information multi-agent settings, knowing the values of states when all agents play an equilibrium is sufficient to reconstruct an equilibrium. However, this does not work in imperfect-information games, as we demonstrate in the next section.

## 2   The Challenge of Depth-Limited Solving in Imperfect-Information Games

In imperfect-information games (also referred to as partially-observable games), an optimal strategy cannot be determined in a subgame simply by knowing the values of states (i.e., game-tree nodes) when all players play an equilibrium strategy. A simple demonstration is in Figure 1a, which shows a sequential game we call Rock-Paper-Scissors+ (RPS+). RPS+ is identical to traditional Rock-Paper-Scissors, except if either player plays Scissors, the winner receives 2 points instead of 1 (and the loser loses 2 points). Figure 1a shows RPS+ as a sequential game in which $P_1$ acts first but does not reveal the action to $P_2$ [7, 13]. The optimal strategy (Minmax strategy, which is also a Nash equilibrium in two-player zero-sum games) for both players in this game is to choose Rock and Paper each with 40% probability, and Scissors with 20% probability. In this equilibrium, the expected value to $P_1$ of choosing Rock is 0, as is the value of choosing Scissors or Paper. In other words, all the red states in Figure 1a have value 0 in the equilibrium. Now suppose $P_1$ conducts a depth-limited search with a depth of one in which the equilibrium values are substituted at that depth limit. This depth-limited subgame is shown in Figure 1b. Clearly, there is not enough information in this subgame to arrive at the optimal strategy of 40%, 40%, and 20% for Rock, Paper, and Scissors, respectively.

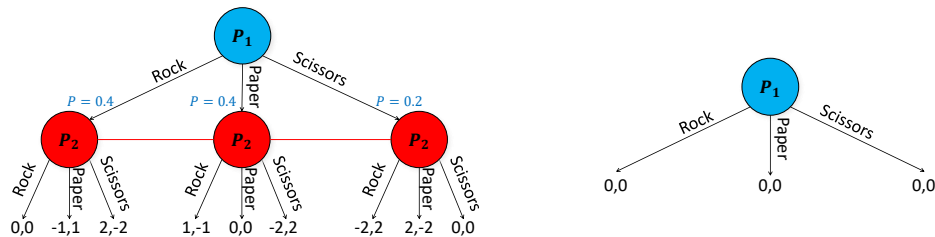

(a) Rock-Paper-Scissors+ shown with the optimal $P_1$ strategy. The terminal values are shown first for $P_1$, then $P_2$. The red lines between the $P_2$ nodes means they are indistinguishable to $P_2$.

(b) A depth-limited subgame of Rock-Paper-Scissors+ with state values determined from the equilibrium.

In the RPS+ example, the core problem is that we incorrectly assumed $P_2$ would always play a fixed strategy. If indeed $P_2$ were to always play Rock, Paper, and Scissors with probability $\langle 0.4, 0.4, 0.2 \rangle$, then $P_1$ could choose any arbitrary strategy and receive an expected value of 0. However, by assuming $P_2$ is playing a fixed strategy, $P_1$ may not find a strategy that is robust to $P_2$ adapting. In reality, $P_2$'s optimal strategy depends on the probability that $P_1$ chooses Rock, Paper, and Scissors. In general, in imperfect-information games a player's optimal strategy at a decision point depends on the player's belief distribution over states as well as the strategy of all other agents beyond that decision point.

In this paper we introduce a method for depth-limited solving that ensures a player is robust to such opponent adaptations. Rather than simply substitute a single state value at a depth limit, we instead allow the opponent one final choice of action at the depth limit, where each action corresponds to a strategy the opponent will play in the remainder of the game. The choice of strategy determines the value of the state. The opponent does not make this choice in a way that is specific to the state (in which case he would trivially choose the maximum value for himself). Instead, naturally, the opponent must make the same choice at all states that are indistinguishable to him. We prove that if the opponent is given a choice between a sufficient number of strategies at the depth limit, then any solution to the depth-limited subgame is part of a Nash equilibrium strategy in the full game. We also show experimentally that when only a few choices are offered (for computational speed), performance of the method is extremely strong.

# 3 Notation and Background

In an imperfect-information extensive-form game there is a finite set of players, $\mathcal{P}$. A state (also called a node) is defined by all information of the current situation, including private knowledge known to only one player. A unique player $P(h)$ acts at state $h$. $H$ is the set of all states in the game tree. The state $h'$ reached after an action is taken in $h$ is a child of $h$, represented by $h \cdot a = h'$, while $h$ is the parent of $h'$. If there exists a sequence of actions from $h$ to $h'$, then $h$ is an ancestor of $h'$ (and $h'$ is a descendant of $h$), represented as $h \sqsubset h'$. $Z \subseteq H$ are terminal states for which no actions are available. For each player $i \in \mathcal{P}$, there is a payoff function $u_i : Z \to \mathbb{R}$. If $P = \{1, 2\}$ and $u_1 = -u_2$, the game is two-player zero-sum. In this paper we assume the game is two-player zero-sum, though many of the ideas extend to general sum and more than two players.

Imperfect information is represented by *information sets* (infosets) for each player $i \in \mathcal{P}$. For any infoset $I$ belonging to player $i$, all states $h, h' \in I$ are indistinguishable to player $i$. Moreover, every non-terminal state $h \in H$ belongs to exactly one infoset for each player $i$.

A strategy $\sigma_i(I)$ (also known as a policy) is a probability vector over actions for player $i$ in infoset $I$. The probability of a particular action $a$ is denoted by $\sigma_i(I, a)$. Since all states in an infoset belonging to player $i$ are indistinguishable, the strategies in each of them must be identical. We define $\sigma_i$ to be a strategy for player $i$ in every infoset in the game where player $i$ acts. A strategy is *pure* if all probabilities in it are 0 or 1. All strategies are a linear combination of pure strategies. A strategy profile $\sigma$ is a tuple of strategies, one for each player. The strategy of every player other than $i$ is represented as $\sigma_{-i}$. $u_i(\sigma_i, \sigma_{-i})$ is the expected payoff for player $i$ if all players play according to the strategy profile $\langle \sigma_i, \sigma_{-i} \rangle$. The value to player $i$ at state $h$ given that all players play according to strategy profile $\sigma$ is defined as $v_i^\sigma(h)$, and the value to player $i$ at infoset $I$ is defined as $v^\sigma(I) = \sum_{h \in I} \left( p(h) v_i^\sigma(h) \right)$, where $p(h)$ is player $i$'s believed probability that they are in state $h$, conditional on being in infoset $I$, based on the other players' strategies and chance's probabilities.

A *best response* to $\sigma_{-i}$ is a strategy $BR(\sigma_{-i})$ such that $u_i(BR(\sigma_{-i}), \sigma_{-i}) = \max_{\sigma_i'} u_i(\sigma_i', \sigma_{-i})$. A *Nash equilibrium* $\sigma^*$ is a strategy profile where every player plays a best response: $\forall i, u_i(\sigma_i^*, \sigma_{-i}^*) = \max_{\sigma_i'} u_i(\sigma_i', \sigma_{-i}^*)$ [29]. A *Nash equilibrium strategy* for player $i$ is a strategy $\sigma_i^*$ that is part of any Nash equilibrium. In two-player zero-sum games, if $\sigma_i$ and $\sigma_{-i}$ are both Nash equilibrium strategies, then $\langle \sigma_i, \sigma_{-i} \rangle$ is a Nash equilibrium.

A *depth-limited imperfect-information subgame*, which we refer to simply as a *subgame*, is a contiguous portion of the game tree that does not divide infosets. Formally, a subgame $S$ is a set of states such that for all $h \in S$, if $h \in I_i$ and $h' \in I_i$ for some player $i$, then $h' \in S$. Moreover, if $x \in S$ and $z \in S$ and $x \sqsubset y \sqsubset z$, then $y \in S$. If $h \in S$ but no descendant of $h$ is in $S$, then $h$ is a *leaf node*. Additionally, the infosets containing $h$ are *leaf infosets*. Finally, if $h \in S$ but no ancestor of $h$ is in $S$, then $h$ is a *root node* and the infosets containing $h$ are *root infosets*.

# 4 Multi-Valued States in Imperfect-Information Games

In this section we describe our new method for depth-limited solving in imperfect-information games, which we refer to as *multi-valued states*. Our general approach is to first precompute an approximate Nash equilibrium for the entire game. We refer to this precomputed strategy profile as a *blueprint* strategy. Since the blueprint is precomputed for the entire game, it is likely just a coarse approximation of a true Nash equilibrium. Our goal is to compute a better approximation in real time for just a depth-limited subgame $S$ that we find ourselves in during play. For the remainder of this paper, we assume that player $P_1$ is attempting to approximate a Nash equilibrium strategy in $S$.

Let $\sigma^*$ be an exact Nash equilibrium. To present the intuition for our approach, we begin by considering what information about $\sigma^*$ would, in theory, be sufficient in order to compute a $P_1$ Nash equilibrium strategy in $S$. For ease of understanding, when considering the intuition for multi-valued states we suggest the reader first focus on the case where $S$ is rooted at the start of the game (that is, no prior actions have occurred).

As explained in Section 2, knowing the values of leaf nodes in $S$ when both players play according to $\sigma^*$ (that is, $v_i^{\sigma^*}(h)$ for leaf node $h$ and player $P_i$) is insufficient to compute a Nash equilibrium in $S$ (even though this is sufficient in perfect-information games), because it assumes $P_2$ would not adapt their strategy outside $S$. But what if $P_2$ could adapt? Specifically, suppose hypothetically that $P_2$

could choose *any* strategy in the entire game, while $P_1$ could only play according to $\sigma_1^*$ outside of $S$. In this case, what strategy should $P_1$ choose in $S$? Since $\sigma_1^*$ is a Nash equilibrium strategy and $P_2$ can choose any strategy in the game (including a best response to $P_1$'s strategy), so by definition $P_1$ cannot do better than playing $\sigma_1^*$ in $S$. Thus, $P_1$ should play $\sigma_1^*$ (or some equally good Nash equilibrium) in $S$.

Another way to describe this setup is that upon reaching a leaf node $h$ in infoset $I$ in subgame $S$, rather than simply substituting $v_2^{\sigma^*}(h)$ (which assumes $P_2$ plays according to $\sigma_2^*$ for the remainder of the game), $P_2$ could instead choose any mixture of pure strategies for the remainder of the game. So if there are $N$ possible pure strategies following $I$, $P_2$ would choose among $N$ actions upon reaching $I$, where action $n$ would correspond to playing pure strategy $\sigma_2^n$ for the remainder of the game. Since this choice is made separately at each infoset $I$ and since $P_2$ may mix between pure strategies, so this allows $P_2$ to choose *any* strategy below $S$.

Since the choice of action would define a $P_2$ strategy for the remainder of the game and since $P_1$ is known to play according to $\sigma_1^*$ outside $S$, so the chosen action could immediately reward the expected value $v_i^{\langle \sigma_1^*, \sigma_2^n \rangle}(h)$ to $P_i$. Therefore, in order to reconstruct a $P_1$ Nash equilibrium in $S$, it is sufficient to know for every leaf node the expected value of every pure $P_2$ strategy against $\sigma_1^*$ (stated formally in Proposition 1). This is in contrast to perfect-information games, in which it is sufficient to know for every leaf node just the expected value of $\sigma_2^*$ against $\sigma_1^*$. Critically, it is not necessary to know the *strategy* $\sigma_1^*$, just the *values* of $\sigma_1^*$ played against every pure opponent strategy in each leaf node.

Proposition 1 adds the condition that we know $v_2^{\langle \sigma_1^*, BR(\sigma_1^*) \rangle}(I)$ for every root infoset $I \in S$. This condition is used if $S$ does not begin at the start of the game. Knowledge of $v_2^{\langle \sigma_1^*, BR(\sigma_1^*) \rangle}(I)$ is needed to ensure that any strategy $\sigma_1$ that $P_1$ computes in $S$ cannot be exploited by $P_2$ changing their strategy earlier in the game. Specifically, we add a constraint that $v_2^{\langle \sigma_1, BR(\sigma_1^*) \rangle}(I) \leq v_2^{\langle \sigma_1^*, BR(\sigma_1^*) \rangle}(I)$ for all $P_2$ root infosets $I$. This makes our technique *safe*:

**Proposition 1.** *Assume $P_1$ has played according to Nash equilibrium strategy $\sigma_1^*$ prior to reaching a depth-limited subgame $S$ of a two-player zero-sum game. In order to calculate the portion of a $P_1$ Nash equilibrium strategy that is in $S$, it is sufficient to know $v_2^{\langle \sigma_1^*, BR(\sigma_1^*) \rangle}(I)$ for every root $P_2$ infoset $I \in S$ and $v_1^{\langle \sigma_1^*, \sigma_2 \rangle}(h)$ for every pure undominated $P_2$ strategy $\sigma_2$ and every leaf node $h \in S$.*

Other safe subgame solving techniques have been developed in recent papers, but those techniques require solving to the end of the full game [7, 17, 28, 5, 6] (except one [27], which we will compare to in Section 7).

Of course, it is impractical to know the expected value in every state of every pure $P_2$ strategy against $\sigma_1^*$, especially since we do not know $\sigma_1^*$ itself. To deal with this, we first compute a blueprint strategy $\hat{\sigma}^*$ (that is, a precomputed approximate Nash equilibrium for the full game). Next, rather than consider every pure $P_2$ strategy, we instead consider just a small number of different $P_2$ strategies (that may or may not be pure). Indeed, in many complex games, the possible opponent strategies at a decision point can be approximately grouped into just a few "meta-strategies", such as which highway lane a car will choose in a driving simulation. In our experiments, we find that excellent performance is obtained in poker with fewer than ten opponent strategies. In part, excellent performance is possible with a small number of strategies because the choice of strategy beyond the depth limit is made separately at each leaf infoset. Thus, if the opponent chooses between ten strategies at the depth limit, but makes this choice independently in each of 100 leaf infosets, then the opponent is actually choosing between $10^{100}$ different strategies. We now consider two questions. First, how do we compute the blueprint strategy $\hat{\sigma}_1^*$? Second, how do we determine the set of $P_2$ strategies? We answer each of these in turn.

There exist several methods for constructing a blueprint. One option, which achieves the best empirical results and is what we use, involves first abstracting the game by bucketing together similar situations [19, 12] and then applying the iterative algorithm Monte Carlo Counterfactual Regret Minimization [22]. Several alternatives exist that do not use a distinct abstraction step [3, 16, 10]. The agent will never actually play according to the blueprint $\hat{\sigma}^*$. It is only used to estimate $v^{\langle \sigma_1^*, \sigma_2 \rangle}(h)$.

We now discuss two different ways to select a set of $P_2$ strategies. Ultimately we would like the set of $P_2$ strategies to contain a diverse set of intelligent strategies the opponent might play, so that $P_1$'s solution in a subgame is robust to possible $P_2$ adaptation. One option is to bias the $P_2$ blueprint

strategy $\hat{\sigma}_2^*$ in a few different ways. For example, in poker the blueprint strategy should be a mixed strategy involving some probability of folding, calling, or raising. We could define a new strategy $\sigma_2'$ in which the probability of folding is multiplied by 10 (and then all the probabilities renormalized). If the blueprint strategy $\hat{\sigma}^*$ were an exact Nash equilibrium, then any such "biased" strategy $\sigma_2'$ in which the probabilities are arbitrarily multiplied would still be a best response to $\hat{\sigma}_1^*$. In our experiments, we use this biasing of the blueprint strategy to construct a set of four opponent strategies on the second betting round. We refer to this as the *bias approach*.

Another option is to construct the set of $P_2$ strategies via self-play. The set begins with just one $P_2$ strategy: the blueprint strategy $\hat{\sigma}_2^*$. We then solve a depth-limited subgame rooted at the start of the game and going to whatever depth is feasible to solve, giving $P_2$ only the choice of this $P_2$ strategy at leaf infosets. That is, at leaf node $h$ we simply substitute $v_i^{\hat{\sigma}^*}(h)$ for $P_i$. Let the $P_1$ solution to this depth-limited subgame be $\sigma_1$. We then approximate a $P_2$ best response assuming $P_1$ plays according to $\sigma_1$ in the depth-limited subgame and according to $\hat{\sigma}_1^*$ in the remainder of the game. Since $P_1$ plays according to this fixed strategy, approximating a $P_2$ best response is equivalent to solving a Markov Decision Process, which is far easier to solve than an imperfect-information game. This $P_2$ approximate best response is added to the set of strategies that $P_2$ may choose at the depth limit, and the depth-limited subgame is solved again. This process repeats until the set of $P_2$ strategies grows to the desired size. This self-generative approach bears some resemblance to the double oracle algorithm [26] and recent work on generation of opponent strategies in multi-agent RL [23]. In our experiments, we use this self-generative method to construct a set of ten opponent strategies on the first betting round. We refer to this as the *self-generative approach*.

One practical consideration is that since $\hat{\sigma}_1^*$ is not an exact Nash equilibrium, a generated $P_2$ strategy $\sigma_2$ may do better than $\hat{\sigma}_2^*$ against $\hat{\sigma}_1^*$. In that case, $P_1$ may play more conservatively than $\sigma_1^*$ in a depth-limited subgame. To correct for this, one can balance the players by also giving $P_1$ a choice between multiple strategies for the remainder of the game at the depth limit. Alternatively, one can "weaken" the generated $P_2$ strategies so that they do no better than $\hat{\sigma}_2^*$ against $\hat{\sigma}_1^*$. Formally, if $v_2^{\langle \hat{\sigma}_1^*, \sigma_2 \rangle}(I) > v_2^{\langle \hat{\sigma}_1^*, \hat{\sigma}_2^* \rangle}(I)$, we uniformly lower $v_2^{\langle \hat{\sigma}_1^*, \sigma_2 \rangle}(h)$ for $h \in I$ by $v_2^{\langle \hat{\sigma}_1^*, \sigma_2 \rangle}(I) - v_2^{\langle \hat{\sigma}_1^*, \hat{\sigma}_2^* \rangle}(I)$. Another alternative (or additional) solution would be to simply reduce $v_2^{\langle \hat{\sigma}_1^*, \sigma_2 \rangle}(h)$ for $\sigma_2 \neq \hat{\sigma}_2^*$ by some heuristic amount, such as a small percentage of the pot in poker.

Once a $P_1$ strategy $\hat{\sigma}_1^*$ and a set of $P_2$ strategies have been generated, we need some way to calculate and store $v_2^{\langle \hat{\sigma}_1^*, \sigma_2 \rangle}(h)$. Calculating the state values can be done by traversing the entire game tree once. However, that may not be feasible in large games. Instead, one can use Monte Carlo simulations to approximate the values. For storage, if the number of states is small (such as in the early part of the game tree), one could simply store the values in a table. More generally, one could train a function to predict the values corresponding to a state, taking as input a description of the state and outputting a value for each $P_2$ strategy. Alternatively, one could simply store $\hat{\sigma}_1^*$ and the set of $P_2$ strategies. Then, in real time, the value of a state could be estimated via Monte Carlo rollouts. We present results for both of these approaches in Section 6.

## 5  Nested Solving of Imperfect-Information Games

We use the new idea discussed in the previous section in the context of nested solving, which is a way to repeatedly solve subgames as play descends down the game tree [5]. Whenever an opponent chooses an action, a subgame is generated following that action. This subgame is solved, and its solution determines the strategy to play until the next opponent action is taken.

Nested solving is particularly useful in dealing with large or continuous action spaces, such as an auction that allows any bid in dollar increments up to $10,000. To make these games feasible to solve, it is common to apply *action abstraction*, in which the game is simplified by considering only a few actions (both for ourselves and for the opponent) in the full action space. For example, an action abstraction might only consider bid increments of $100. However, if the opponent chooses an action that is not in the action abstraction (called an *off-tree action*), the optimal response to that opponent action is undefined.

Prior to the introduction of nested solving, it was standard to simply round off-tree actions to a nearby in-abstraction action (such as treating an opponent bid of $150 as a bid of $200) [14, 34, 11]. Nested solving allows a response to be calculated for off-tree actions by constructing and solving a subgame

that immediately follows that action. The goal is to find a strategy in the subgame that makes the opponent no better off for having chosen the off-tree action than an action already in the abstraction.

Depth-limited solving makes nested solving feasible even in the early game, so it is possible to play without acting according to a precomputed strategy or using action translation. At the start of the game, we solve a depth-limited subgame (using action abstraction) to whatever depth is feasible. This determines our first action. After every opponent action, we solve a new depth-limited subgame that attempts to make the opponent no better off for having chosen that action than an action that was in our previous subgame's action abstraction. This new subgame determines our next action, and so on.

## 6 Experiments

We conducted experiments on the games of heads-up no-limit Texas hold'em poker (HUNL) and heads-up no-limit flop hold'em poker (NLFH). Appendix B reminds the reader of the rules of these games. HUNL is the main large-scale benchmark for imperfect-information game AIs. NLFH is similar to HUNL, except the game ends immediately after the second betting round, which makes it small enough to precisely calculate best responses and Nash equilibria. Performance is measured in terms of mbb/g, which is a standard win rate measure in the literature. It stands for milli-big blinds per game and represents how many thousandths of a big blind (the initial money a player must commit to the pot) a player wins on average per hand of poker played.

### 6.1 Exploitability Experiments in No-Limit Flop Hold'em (NLFH)

Our first experiment measured the *exploitability* of our technique in NLFH. Exploitability of a strategy in a two-player zero-sum game is how much worse the strategy would do against a best response than a Nash equilibrium strategy would do against a best response. Formally, the exploitability of $\sigma_1$ is $\min_{\sigma_2} u_1(\sigma_1^*, \sigma_2) - \min_{\sigma_2} u_1(\sigma_1, \sigma_2)$, where $\sigma_1^*$ is a Nash equilibrium strategy.

We considered the case of $P_1$ betting $0.75\times$ the pot at the start of the game, when the action abstraction only contains bets of $0.5\times$ and $1\times$ the pot. We compared our depth-limited solving technique to the randomized pseudoharmonic action translation (RPAT) [11], in which the bet of $0.75\times$ is simply treated as either a bet of $0.5\times$ or $1\times$. RPAT is the lowest-exploitability known technique for responding to off-tree actions that does not involve real-time computation.

We began by calculating an approximate Nash equilibrium in an action abstraction that does not include the $0.75\times$ bet. This was done by running the CFR+ equilibrium-approximation algorithm [38] for 1,000 iterations, which resulted in less than 1 mbb/g of exploitability within the action abstraction. Next, values for the states at the end of the first betting round within the action abstraction were determined using the self-generative method discussed in Section 4. Since the first betting round is a small portion of the entire game, storing a value for each state in a table required just 42 MB.

To determine a $P_2$ strategy in response to the $0.75\times$ bet, we constructed a depth-limited subgame rooted after the $0.75\times$ bet with leaf nodes at the end of the first betting round. The values of a leaf node in this subgame were set by first determining the in-abstraction leaf nodes corresponding to the exact same sequence of actions, except $P_1$ initially bets $0.5\times$ or $1\times$ the pot. The leaf node values in the $0.75\times$ subgame were set to the average of those two corresponding value vectors. When the end of the first betting round was reached and the board cards were dealt, the remaining game was solved using safe subgame solving.

Figure 2 shows how exploitability decreases as we add state values (that is, as we give $P_1$ more best responses to choose from at the depth limit). When using only one state value at the depth limit (that is, assuming $P_1$ would always play according to the blueprint strategy for the remainder of the game), it is actually better to use RPAT. However, after that our technique becomes significantly better and at 16 values its performance is close to having had the $0.75\times$ action in the abstraction in the first place.

While one could have calculated a (slightly better) $P_2$ strategy in response to the $0.75\times$ bet by solving to the end of the game, that subgame would have been about $10,000\times$ larger than the subgames solved in this experiment. Thus, depth-limited solving dramatically reduces the computational cost of nested subgame solving while giving up very little solution quality.

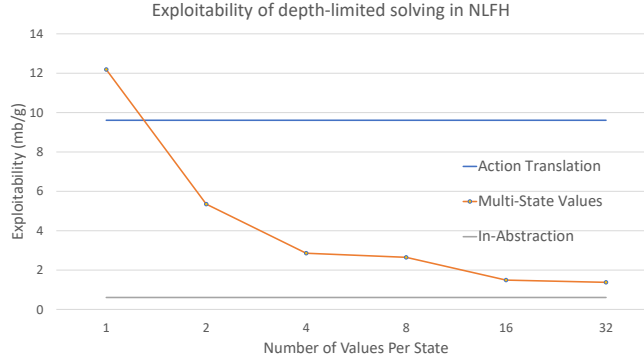

Figure 2: Exploitability of depth-limited solving in response to an opponent off-tree action as a function of number of state values. We compare to action translation and to having had the off-tree action included in the action abstraction (which is a lower bound on the exploitability achievable with 1,000 iterations of CFR+).

## 6.2 Experiments Against Top AIs in Heads-Up No-Limit Texas Hold'em (HUNL)

Our main experiment uses depth-limited solving to produce a master-level HUNL poker AI called *Modicum* using computing resources found in a typical laptop. We test *Modicum* against Baby Tartanian8 [4], the winner of the 2016 Annual Computer Poker Competition, and against Slumbot [18], the winner of the 2018 Annual Computer Poker Competition. Neither Baby Tartanian8 nor Slumbot uses real time computation; their strategies are a precomputed lookup table. Baby Tartanian8 used about 2 million core hours and 18 TB of RAM to compute its strategy. Slumbot used about 250,000 core hours and 2 TB of RAM to compute its strategy. In contrast, *Modicum* used just 700 core hours and 16GB of RAM to compute its strategy and can play in real time at the speed of human professionals (an average of 20 seconds for an entire hand of poker) using just a 4-core CPU. We now describe *Modicum* and provide details of its construction in Appendix A.

The blueprint strategy for *Modicum* was constructed by first generating an abstraction of HUNL using state-of-the-art abstraction techniques [12, 20]. Storing a strategy for this abstraction as 4-byte floats requires just 5 GB. This abstraction was approximately solved by running Monte Carlo Counterfactual Regret Minimization for 700 core hours [22].

HUNL consists of four betting rounds. We conduct depth-limited solving on the first two rounds by solving to the end of that round using MCCFR. Once the third betting round is reached, the remaining game is small enough that we solve to the end of the game using an enhanced form of CFR+ described in the appendix.

We generated 10 values for each state at the end of the first betting round using the self-generative approach. The first betting round was small enough to store all of these state values in a table using 240 MB. For the second betting round, we used the bias approach to generate four opponent best responses. The first best response is simply the opponent's blueprint strategy. For the second, we biased the opponent's blueprint strategy toward folding by multiplying the probability of fold actions by 10 and then renormalizing. For the third, we biased the opponent's blueprint strategy toward checking and calling. Finally for the fourth, we biased the opponent's blueprint strategy toward betting and raising. To estimate the values of a state when the depth limit is reached on the second round, we sample rollouts of each of the stored best-response strategies.

The performance of *Modicum* is shown in Table 1. For the evaluation, we used AIVAT to reduce variance [8]. Our new agent defeats both Baby Tartanian8 and Slumbot with statistical significance. For comparison, Baby Tartanian8 defeated Slumbot by $36 \pm 12$ mbb/g, *Libratus* defeated Baby Tartanian8 by $63 \pm 28$ mbb/g, and *Libratus* defeated top human professionals by $147 \pm 77$ mbb/g.

In addition to head-to-head performance against prior top AIs, we also tested *Modicum* against two versions of *Local Best Response (LBR)* [25]. An LBR agent is given full access to its opponent's full-game strategy and uses that knowledge to exactly calculate the probability the LBR agent is in each possible state. Given that probability distribution and a heuristic for how the opposing agent will play thereafter, the LBR agent chooses a best response action. LBR is a way to calculate a lower bound on exploitability and has been shown to be effective in exploiting agents that do not use real-time solving.

|  | Baby Tartanian8 | Slumbot |
|---|---|---|
| Blueprint (No real-time solving) | $-57 \pm 13$ | $-11 \pm 8$ |
| Naïve depth-limited solving | $-10 \pm 8$ | $-1 \pm 15$ |
| Depth-limited solving | $6 \pm 5$ | $11 \pm 9$ |

Table 1: Head to head performance of our new agent against Baby Tartanian8 and Slumbot with 95% confidence intervals shown. Our new agent defeats both opponents with statistical significance. Naïve depth-limited solving means states are assumed to have just a single value, which is determined by the blueprint strategy.

In the first version of LBR we tested against, the LBR agent was limited to either folding or betting $0.75\times$ the pot on the first action, and thereafter was limited to either folding or calling. *Modicum* beat this version of LBR by $570 \pm 42$ mbb/g. The second version of LBR we tested against could bet 10 different amounts on the flop that *Modicum* did not include in its blueprint strategy. Much like the experiment in Section 6.1, this was intended to measure how vulnerable *Modicum* is to unanticipated bet sizes. The LBR agent was limited to betting $0.75\times$ the pot for the first action of the game and calling for the remaining actions on the preflop. On the flop, the LBR agent could either fold, call, or bet $0.33 \times 2^x$ times the pot for $x \in \{0, 1, ..., 10\}$. On the remaining rounds the LBR agent could either fold or call. *Modicum* beat this version of LBR by $1377 \pm 115$ mbb/g. In contrast, similar forms of LBR have been shown to defeat prior top poker AIs that do not use real-time solving by hundreds or thousands of mbb/g [25].

While our new agent is probably not as strong as *Libratus*, it was produced with less than $0.1\%$ of the computing resources and memory, and is never vulnerable to off-tree opponent actions.

While the rollout method used on the second betting round worked well, rollouts may be significantly more expensive in deeper games. To demonstrate the generality of our approach, we also trained a deep neural network (DNN) to predict the values of states at the end of the second betting round as an alternative to using rollouts. The DNN takes as input a 34-float vector of features describing the state, and outputs four floats representing the values of the state for the four possible opponent strategies (represented as a fraction of the size of the pot). The DNN was trained using 180 million examples per player by optimizing the Huber loss with Adam [21], which we implemented using PyTorch [32]. In order for the network to run sufficiently fast on just a 4-core CPU, the DNN has just 4 hidden layers with 256 nodes in the first hidden layer and 128 nodes in the remaining hidden layers. This achieved a Huber loss of 0.02. Using a DNN rather than rollouts resulted in the agent beating Baby Tartanian8 by $2 \pm 9$ mbb/g. However, the average time taken using a 4-core CPU increased from 20 seconds to 31 seconds per hand. Still, these results demonstrate the generality of our approach.

# 7  Comparison to Prior Work

Section 2 demonstrated that in imperfect-information games, states do not have unique values and therefore the techniques common in perfect-information games and single-agent settings do not apply. This paper introduced a way to overcome this challenge by assigning multiple values to states. A different approach is to modify the definition of a "state" to instead be all players' belief probability distributions over states, which we refer to as a *joint belief state*. This technique was previously used to develop the poker AI *DeepStack* [27]. While *DeepStack* defeated non-elite human professionals in HUNL, it was never shown to defeat prior top AIs even though it used over 1,000,000 core hours of computation. In contrast, *Modicum* defeated two prior top AIs with less than 1,000 core hours of computation. Still, there are benefits and drawbacks to both approaches, which we now describe in detail. The right choice may depend on the domain and future research may change the competitiveness of either approach.

A joint belief state is defined by a probability (belief) distribution for each player over states that are indistinguishable to the player. In poker, for example, a joint belief state is defined by each players' belief about what cards the other players are holding. Joint belief states maintain some of the properties that regular states have in perfect-information games. In particular, it is possible to determine an optimal strategy in a subgame rooted at a joint belief state independently from the rest of the game. Therefore, joint belief states have unique, well-defined values that are not influenced by the strategies played in disjoint portions of the game tree. Given a joint belief state, it is also possible

to define the value of each root infoset for each player. In the example of poker, this would be the value of a player holding a particular poker hand given the joint belief state.

One way to do depth-limited subgame solving, other than the method we describe in this paper, is to learn a function that maps joint belief states to infoset values. When conducting depth-limited solving, one could then set the value of a leaf infoset based on the joint belief state at that leaf infoset.

One drawback is that because a player's belief distribution partly defines a joint belief state, the values of the leaf infosets must be recalculated each time the strategy in the subgame changes. With the best domain-specific iterative algorithms, this would require recalculating the leaf infosets about 500 times. Monte Carlo algorithms, which are the preferred domain-independent method of solving imperfect-information games, may change the strategy millions of times in a subgame, which poses a problem for the joint belief state approach. In contrast, our multi-valued state approach requires only a single function call for each leaf node regardless of the number of iterations conducted.

Moreover, evaluating multi-valued states with a function approximator is cheaper and more scalable to large games than joint belief states. The input to a function that predicts the value of a multi-valued state is simply the state description (for example, the sequence of actions), and the output is several values. In our experiments, the input was 34 floats and the output was 4 floats. In contrast, the input to a function that predicts the values of a joint belief state is a probability vector for each player over the possible states they may be in. For example, in HUNL, the input is more than 2,000 floats and the output is more than 1,000 floats. The input would be even larger in games with more states per infoset.

Another drawback is that learning a mapping from joint belief states to infoset values is computation-ally more expensive than learning a mapping from states to a set of values. For example, *Modicum* required less than 1,000 core hours to create this mapping. In contrast, *DeepStack* required over 1,000,000 core hours to create its mapping. The increased cost is partly because computing training data for a joint belief state value mapping is inherently more expensive. The multi-valued states approach is learning the values of *best responses* to a particular strategy (namely, the approximate Nash equilibrium strategy $\hat{\sigma}_1^*$). In contrast, a joint belief state value mapping is learning the value of all players playing an *equilibrium* strategy given that joint belief state. As a rough guideline, computing an equilibrium is about $1,000\times$ more expensive than computing a best response in large games [1].

On the other hand, the multi-valued state approach requires knowledge of a blueprint strategy that is already an approximate Nash equilibrium. A benefit of the joint belief state approach is that rather than simply learning best responses to a particular strategy, it is learning best responses against *every possible* strategy. This may be particularly useful in self-play settings where the blueprint strategy is unknown, because it may lead to increasingly more sophisticated strategies.

Another benefit of the joint belief state approach is that in many games (but not all) it obviates the need to keep track of the sequence of actions played. For example, in poker if there are two different sequences of actions that result in the same amount of money in the pot and all players having the same belief distribution over what their opponents' cards are, then the optimal strategy in both of those situations is the same. This is similar to how in Go it is not necessary to know the exact sequence of actions that were played. Rather, it is only necessary to know the current configuration of the board (and, in certain situations, also the last few actions played).

A further benefit of the joint belief state approach is that its run-time complexity does not increase with the degree of precision other than needing a better (possibly more computationally expensive) function approximator. In contrast, for our algorithm the computational complexity of finding a solution to a depth-limited subgame grows linearly with the number of values per state.

## 8   Conclusions

We introduced a principled method for conducting depth-limited solving in imperfect-information games. Experimental results show that this leads to stronger performance than the best precomputed-strategy AIs in HUNL while using orders of magnitude less computational resources, and is also orders of magnitude more efficient than past approaches that use real-time solving. Additionally, the method exhibits low exploitability. In addition to using less resources, this approach broadens the applicability of nested real-time solving to longer games.

# 9 Acknowledgments

This material is based on work supported by the National Science Foundation under grants IIS-1718457, IIS-1617590, and CCF-1733556, and the ARO under award W911NF-17-1-0082, as well as XSEDE computing resources provided by the Pittsburgh Supercomputing Center. We thank Thore Graepel, Marc Lanctot, David Silver, Ariel Procaccia, Fei Fang, and our anonymous reviewers for helpful inspiration, feedback, suggestions, and support.

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
