[Supplementary Material · supplementary.pdf]

# Appendix: Supplementary Material

## A   Details of How We Constructed the *Modicum* Agent

In this section we provide details on the construction of our new agent and the implementation of depth-limited subgame solving, as well as a number of optimizations we used to improve the performance of our agent.

The blueprint abstraction treats every poker hand separately on the first betting round (where there are 169 strategically distinct hands). On the remaining betting rounds, the hands are grouped into 30,000 buckets [2, 12, 20]. The hands in each bucket are treated identically and have a shared strategy, so they can be thought as sharing an *abstract infoset*. The action abstraction was chosen primarily by observing the most common actions used by prior top agents. We made a conscious effort to avoid actions that would likely not be in Baby Tartanian8's and Slumbot's action abstraction, so that we do not actively exploit their use of action translation. This makes our experimental results relatively conservative. While we do not play according to the blueprint strategy, the blueprint strategy is nevertheless used to estimate the values of states, as explained in the body of the paper.

We used *unsafe* nested solving on the first and second betting rounds, as well as for the first subgame on the third betting round. In unsafe solving [13], each player maintains a belief distribution over states. When the opponent takes an action, that belief distribution is updated via Bayes' rule assuming that the opponent played according to the equilibrium we had computed. Unsafe solving lacks theoretical guarantees because the opponent need not play according to the specific equilibrium we compute, and may actively exploit our assumption that they are playing according to a specific strategy. Nevertheless, in practice unsafe solving typically achieves strong performance and exhibits low exploitability, particularly in large games [5].

In nested unsafe solving, whenever the opponent chooses an action, we generate a subgame rooted immediately *before* that action was taken (that is, the subgame starts with the opponent acting). The opponent is given a choice between actions that we already had in our action abstraction, as well as the new action that they actually took. This subgame is solved (in our case, using depth-limited solving). The solution's probability for the action the opponent actually took informs how we update the belief distribution of the other player. The solution also gives a strategy for the player who now acts. This process repeats each time the opponent acts.

Since the first betting round (called the *preflop*) is extremely small, whenever the opponent takes an action that we have not previously observed, we add it to the action abstraction for the preflop, solve the whole preflop again, and cache the solution. When the opponent chooses an action that they have taken in the past, we simply load the cached solution rather than solve the subgame again. This results in the preflop taking a negligible amount of time on average.

To determine the values of leaf nodes on the first and second betting round, whenever a subgame was constructed we mapped each leaf node in the subgame to a leaf node in the blueprint abstraction (based on similarity of the action sequence). The values of a leaf node in the subgame (as a fraction of the pot) was set to its corresponding blueprint abstraction leaf node. In the case of rollouts, this meant conducting rollouts in the blueprint strategy starting at the blueprint leaf node.

As explain in the body of the paper, we tried two methods for determining state values at the end of the second betting round. The first method involves storing the four opponent approximate best responses and doing rollouts in real time whenever the depth limit is reached. The second involves training a deep neural network (DNN) to predict the state values determined by the four approximate best responses.

For the rollout method, it is not necessary to store the best responses as 4-byte floats. That would use $32|A|$ bits per abstract infoset, where $|A|$ is the number of actions in an infoset. If one is constrained by memory, an option is to randomize over the actions in an abstract infoset ahead of time and pick a single action. That single action can then be stored using a minimal number of bits. This means using only $\lceil log_2(|A|) \rceil$ bits per infoset. This comes at a slight cost of precision, particularly if the strategy is small, because it would mean always picking the same action in an infoset whenever it is sampled. Since we were not severely memory constrained, we instead stored the approximate best responses using a single byte per abstract infoset action. In order to reduce variance and converge

more quickly, we conduct multiple rollouts upon reaching a leaf node. We found the optimal number of rollouts to be three given our memory access speeds.

For the DNN approach, whenever a subgame on the second round is generated we evaluate each leaf node using the DNN before solving begins. The state values are stored (using about 50 MB). This takes between 5 and 10 seconds depending on the size of the subgame.

Starting on the third betting round, we always solve to the end of the game using an improved form of CFR+. We use unsafe solving the first time the third betting round is reached. Subsequent subgames are solved using safe nested solving (specifically, Reach subgame solving where the alternative payoffs are based on the expected value from the previously-solved subgame [5]).

To improve the performance of CFR+, we ignore the first 50% of iterations when determining the average strategy. Moreover, for the first 30 iterations, we discount the regrets after each iteration by $\frac{\sqrt{T}}{\sqrt{T+1}}$ where $T$ indicates the iteration. This reduces exploitability in the subgame by about a factor of three.

The number of CFR+ iterations and the amount of time we ran MCCFR varied depending on the size of the pot. For the preflop, we always ran MCCFR for 30 seconds to solve a subgame (though this was rarely done due to caching). On the flop, we ran MCCFR for 10 to 30 seconds depending on the pot size. On the turn, we ran between 150 and 1,000 iterations of our modified form CFR+. On the river, we ran between 300 and 2,000 iterations of our modified form of CFR+.

# B Rules of the Poker Games

We experiment on two variants of poker: heads-up no-limit Texas hold'em (HUNL) and heads-up no-limit flop hold'em (NLFH).

In the version of HUNL we use in this paper, and which is standard in the Annual Computer Poker Competition, the two players ($P_1$ and $P_2$) in the game start each hand with $20,000. The position of the two players alternate after each hand. There are four rounds of betting. On each round, each player can choose to either fold, call, or raise. Folding results in the player losing and the money in the pot being awarded to the other player. Calling means the player places a number of chips in the pot equal to the opponent's share. Raising means that player adds more chips to the pot than the opponent's share. A round ends when a player calls (if both players have acted). Players cannot raise beyond the $20,000 they start with, so there is a limited number of actions in the game. All raises must be at least $100, and at least as larger as the most recent raise on that round (if there was one).

At the start of each hand of HUNL, both players are dealt two private cards from a standard 52-card deck. $P_1$ must place $100 in the pot and $P_2$ must place $50 in the pot. A round of betting then occurs. When the round ends, three *community* cards are dealt face up that both players can ultimately use in their final hands. Another round of betting occurs, starting with $P_1$ this time. After the round is over, another community card is dealt face up, and another round of betting starts with $P_1$ acting first. Finally, one more community card is revealed to both players and a final betting round occurs starting with $P_1$. Unless a player has folded, the player with the best five-card poker hand, constructed from their two private cards and the five community cards, wins the pot. In the case of a tie, the pot is split evenly.

NLFH is similar to HUNL except there are only two rounds of betting and three community cards.

# C Proof of Proposition 1

*Proof.* Consider the augmented subgame $S'$ structured as follows. $S'$ contains $S$ and all its descendants. Additionally, for every root node $h \in S$ (that is, a node whose parent is not in $S$), $S'$ contains a node $h'$ belonging to $P_2$. If $h_1$ and $h_2$ are root nodes in $S$ and $h_1$ and $h_2$ share an infoset, then $h'_1$ and $h'_2$ share an infoset. $S'$ begins with an initial chance node that reaches $h'$ with probability proportional to the probability of reaching $h$ if $P_2$ tried to do so (that is, the probability of reaching it according to $P_1$'s strategy and chance's probabilities).

At node $h'$, $P_2$ has two actions. The "alt" action leads to a terminal node that awards $v_2^{\langle \sigma_1^*, BR(\sigma_1^*) \rangle}(I)$. The "enter" action leads to $h$. From Theorem 1 in [7], a solution to $S'$ is part of a $P_1$ Nash equilibrium strategy in the full game.

Now consider the depth-limited augmented subgame $S''$ that is similar to $S'$ but does not contain the descendants of $S$. We show that knowing $v_1^{\langle \sigma_1^*, \sigma_2 \rangle}(h)$ for every pure undominated $P_2$ strategy $\sigma_2$ and every leaf node $h \in S$ is sufficient to calculate the portion of a $P_1$ Nash equilibrium strategy for $S'$ that is in $S''$. That, in turn, gives a strategy in $S$ that is a Nash equilibrium strategy in the full game.

We modify $S''$ so that, after $P_1$'s strategy is chosen, $P_2$ chooses a probability distribution over the $N$ pure undominated strategies where the probability of pure undominated strategy $\sigma_2^n$ is represented as $p(n)$. This mixture of pure strategies defines a strategy $\sigma_2^m = \sum_{n \leq N} \left( p(n) \sigma_2^n \right)$. In this way, $P_2$ can pick any undominated strategy because every undominated strategy is a mixture of pure undominated strategies. Upon reaching a leaf node $h$, $P_1$ receives a reward of $\sum_{n \leq N} \left( p(n) v_1^{\langle \sigma_1^*, \sigma_2^n \rangle}(h) \right) = v_1^{\langle \sigma_1^*, \sigma_2^m \rangle}(h)$. Clearly $P_1$ can do no better than playing $\sigma_1^*$, because it is a Nash equilibrium and $P_2$ can play any undominated strategy. Thus, any strategy $P_1$ plays in $S''$, when combined with $\sigma_1^*$ outside of $S''$, must do at least as well as playing $\sigma_1^*$ in the full game. $\qquad\square$