[Reviews · NeurIPS 2018]

Reviewer 1



This paper presents a method for finding approximate Nash equilibria in two player zero-sum games with imperfect information (using poker as the motivating example) using a depth-limited solving technique. Depth-limited techniques avoid the need to traverse an entire game tree by replacing nodes in the tree with an estimate of the expected value of their corresponding subtree (for given strategies). This approach doesn’t work in imperfect information games because optimal strategies can’t be reconstructed from values (this idea is clearly demonstrated in a simple rock-paper-scissors example in Section 2 of the paper). This paper presents a technique for avoiding this problem by building player 1’s strategy in a depth-limited sub-game with respect to a set of candidate player 2 strategies (and their corresponding values) that define how player 2 may play in the subgame below the depth limit. It addresses the lack of depth-limited solving in Libratus [Brown & Sandholm 2017] which necessitated a pre-computated strategy that was extremely computationally expensive and presents an alternative to DeepStack’s [Moravčík et al 2017] depth-limited solving technique. # Quality The key idea of this paper is that you can reconstruct player 1’s Nash strategy with depth-limited solving if at the leaves of a subgame you store player 1’s value for every pure player 2 strategy (as summarized by Proposition 1). This proposition doesn’t appear immediately useful: it assumes we already have player 1’s strategy and that the set of all pure strategies for player 2 grows at the same rate as the full search space, so it doesn’t seem helpful to use depth limiting if you need to store an exponentially large object as every node, but the authors find that it is sufficient to instead use only a constant number of well-chosen strategies with an approximation of player 1’s Nash strategy and still get strong performance. Pros - empirically they’re able to get very good performance with very constrained resources. This is the strongest support of the claim that a small number of strategies is sufficient - there is potentially a large design space in choosing P2’s strategies. I suspect there will be a lot of future work in this area Cons - despite presenting two approaches for building the set of P2 strategies (bias and self-generating) the paper gives very little guidance on what constitutes a good set of strategies. I would expect performance to be very sensitive to your choice of strategies, but this isn’t evaluated. The implementation used in the experiments use the self generative approach for the first betting round and the biased approach thereafter. Why? What happens if you just used the self-generated approach or the biased approach? - The computational restriction to a 4-core CPU and 16 GB of memory makes for a flashy abstract; but I really would have liked to see what happens with a less arbitrarily constrained model. Do improvements in performance saturate as you keep increasing the size of the set of P2 strategies? What does the DNN performance look like if you used the largest model that you could fit on the GPU it was trained on instead of limiting it for fast CPU performance? How does the quality of the blueprint strategy affect performance? All of these questions could easily have been answered without needing massive compute resources, so it seems weird to limit oneself to the resources of a MacBook Air. # Clarity Overall I thought the paper was very clearly written and I enjoyed the simple examples shown in section 2 to describe the problems associated with depth limited search. There were a couple of ambiguous / misleading / etc. statements that could be improved: - lines 147 - 150: the claim around 10^100 strategies seems very misleading. Surely what matters is not the size of the set of the potential number of independent choices, but the coverage of the subset you choose? Unless you can prove something about how the number of independent strategies relates to the quality of the solution I would remove this line. - line 166 “Thus \sigma_2’ certainly qualifies as an intelligent strategy to play” - by that definition any arbitrary strategies is “intelligent” because it has expected value zero against a Nash strategy. I’d argue that’s just a property of a Nash equilibrium not about how intelligent the strategy is… - line 358-359: I don’t think there’s a relationship between the computational complexity of computing a function (such as best response or equilibrium) and the difficulty in learning it (given the same amount of training examples)? Function approximation quality depends on the properties of the function (Lipschitz constants, etc.) rather than how expensive it is to collect the data points. # Significance Despite my reservations about the empirical evaluation of this work, I do this that it is significant. I suspect a number of authors will explore the space of heuristics and learning algorithms for generating a useful set of P2 strategies. It is for this reason that I am recommending its acceptance. But I do think it would have been a far stronger paper if the implementation details were more explicitly evaluated. Typos: - line 190 - should be “\sigma_1^*” not “\sigma_2^*” - line 118 - delete “so” in “so clearly P1 cannot do better”

Reviewer 2



Main ideas ========== This paper proposes a method for solving two player games with imperfect information. The technique used is to perfrom a depth limited game tree search, and at the depth-limit consider a range of values - possible outcomes of following a fixed strategy - and then checking its relation to the best response. The experimental results show that the resulting solver outperforms state of the art Poker AI players, and requires significantly lower computation resources. The authors also experimented with a deep learning based approach for the estimation part of their algorithm, with so-so results. Weakness ========== Clarity: Up to section 4, the paper is clear. From section 4 and onwards, I found it very difficult to understand, because it is not clear what are the simplifying assumptions being made, what are implementation details, and what are the key technical contributions. Also, the authors focus on two player games, but in several places in the paper go back to the more general n-player definitions. This is very confusing and unhelpful. For examples, see lines 94 and 125. Strengh ========= Significance: It seems to me that Proposition 1 is key to the theoretical correctness of the proposed approach, but it is not given a formal proof. Also, the lack of clarity of the proposed method will make it very hard to verify or understand the main ideas underlying this work. However, the state of the art results are extremely impressive. Originality: Novel as far as I could tell. Others: Line 94: \Sigme_i was not defined. Line 145: " ... of three lanes to choose ..." - This sentence is not clear to me at all.

Reviewer 3



Summary: The paper proposes a modification of the continual resolving algorithm for depth-limited search in imperfect information games. Instead of counterfactual value network mapping joint beliefs to counterfactual values, it suggests using a function that maps a public state to a set of values of diverse best response strategies again an approximate Nash equilibrium strategy. This function can be precomputed, or realized either by function approximation of by Monte Carlo simulation based on pre-computed strategies. The authors show that this approach performs well in poker with small computational requirements. Quality: The core ideas of the paper are correct and interesting. There are several technical details that are unclear or overstated. 1) The most important problem are the inconsistencies regarding the blueprint strategy. The paper assumes at several places that the player plays based on a precomputed strategy \sigma^*_1. At the same time, the paper says that the player never plays based on the blueprint strategy. Since the blueprint strategy is created using abstraction, which leads to huge exploitability in HUNL, it is likely substantially different from the strategy computed and played online, especially when the opponent uses actions not included in the abstraction. All the theoretical guarantees relay on the player playing based on the blueprint strategy. Therefore, they make only a very limited sense and serve only as an inspiration for the heuristic approach actually evaluated in the paper. This is not clearly stated. I do not see how the theory can be useful unless the blueprint strategy already solves the game and therefore, there is no need for online computation. I ask the authors to elaborate on this in the rebuttal. 2) The second important weakness of the paper is the lack of experimental comparison with the state of the art. The paper spends whole page explaining reasons why the presented approach might perform better under some circumstances, but there is no hard evidence at all. What is the reason not to perform an empirical comparison to the joint belief state approach and show the real impact of the claimed advantages and disadvantages? Since this is the main point of the paper, it should be clear when the new modification is useful. 3) Furthermore, there is an incorrect statement about the performance of the state of the art method. The paper claims that "The evidence suggests that in the domain we tested on, using multi-valued states leads to better performance." because the alternative approach "was never shown to defeat prior top AIs". This is simply incorrect. Lack of an experiment is not evidence for superiority of the method that performed the experiment without any comparison. 4) The rock-paper-scissors example is clearly inspired by an example that appeared in many previous work. Please, cite the source appropriately. 5) As explained in 1), the presented method is quite heuristic. The algorithm does not actually play the blueprint strategy, only few values are used in the leaf states, which cannot cover the whole variety of the best response values. In order to assess whether the presented approach might be applicable also for other games, it would be very useful to evaluate it on some substantially different domains, besides poker. Clarity: The paper is well written and organized, and it is reasonably easy to understand. The impact of the key differences between the theoretic inspiration and the practical implementation should be explained more clearly. Originality: The presented method is a novel modification of continual resolving. The paper clearly explains the main distinction form the existing method. Significance: The presented method seems to substantially reduce the computational requirements of creating a strong poker bot. If this proofs to be the case also for some other imperfect information games, it would be a very significant advancement in creating algorithms for playing these games. Detailed comments: 190: I guess the index should be 1 339: I would not say MCCFR is currently the preferred solution method, since CFR+ does not work well with sampling 349: There is no evidence the presented method would work better in stratego. It would depend on the specific representation and how well would the NN generalize over the types of heuristics. Reaction to rebuttal: 1) The formulation of the formal statement should be clearer. Still, while you are using the BR values from the blueprint strategy in the computation, I do not see how the theory can give you any real bounds the way you use the algorithm. One way to get more realistic bounds would be to analyze the function approximation version and use error estimates from cross-valiadation. 2) I do not believe head-to-head evaluation makes too much sense because of well known intransitivity effects. However, since the key difference between your algorithm and DeepStack is the form of the used leaf evaluation function, it would certainly not take man-years to replace the evaluation function with the joint belief in your framework. It would be very interesting to see comparison of exploitability and other trade-offs on smaller games, where we can still compute it. 4) I meant the use of the example for save resolving. 5) There is no need for strong agents for some particular games to make rigorous evaluation of equilibrium solving algorithms. You can compute exploitability in sufficiently large games to evaluate how close your approach is to the equilibrium. Furthermore, there are many domain independent algorithms for approaximating equilibriua in these games you can compare to. Especially the small number of best response values necessary for the presented approach is something that would be very interesting to evaluate in other games. Line 339: I just meant that I consider CFR+ to be "the preferred domain-independent method of solving imperfect-information games", but it is not really important, it was a detailed comment.